# Serum-Free Media Formulation Using Marine Microalgae Extracts and Growth Factor Cocktails for Madin-Darby Canine Kidney and Vero Cell Cultures

**DOI:** 10.3390/ijms25189881

**Published:** 2024-09-12

**Authors:** Areumi Park, Yeon-Ji Lee, Eunyoung Jo, Gun-Hoo Park, Seong-Yeong Heo, Eun-Jeong Koh, Seung-Hong Lee, Seon-Heui Cha, Soo-Jin Heo

**Affiliations:** 1Jeju Bio Research Center, Korea Institute of Ocean Science and Technology (KIOST), Jeju 63349, Republic of Korea; areumi1001@kiost.ac.kr (A.P.); leeyj0409@kiost.ac.kr (Y.-J.L.); jey8574@kiost.ac.kr (E.J.); gunhoopark@kiost.ac.kr (G.-H.P.); syheo@kiost.ac.kr (S.-Y.H.); kej763@kiost.ac.kr (E.-J.K.); 2Department of Marine Technology & Convergence Engineering (Marine Biotechnology), University of Science and Technology (UST), Daejeon 34113, Republic of Korea; 3Department of Pharmaceutical Engineering, Soonchunhyang University, Asan 31538, Republic of Korea; shlee80@sch.ac.kr; 4Department of Marine Bio and Medical Sciences, Hanseo University, Seosan-si 32158, Republic of Korea; sunnycha@hanseo.ac.kr

**Keywords:** marine microalgae, extracts, serum-free media, MDCK cells, Vero cells, cell proliferation, antioxidant activity

## Abstract

The development of serum-free media (SFM) is critical to advance cell culture techniques used in viral vaccine production and address the ethical concerns and contamination risks associated with fetal bovine serum (FBS). This study evaluated the effects of marine microalgal extracts and growth factor cocktails on the activity of Madin-Darby canine kidney (MDCK) and Vero cells. Five marine microalgal species were used: *Spirulina platensis* (SP), *Dunaliella salina* (DS), *Haematococcus pluvialis* (HP), *Nannochloropsis salina* (NS), and *Tetraselmis* sp. (TS). DS and SP extracts significantly increased the proliferation rate of both MDCK and Vero cells. DS had a proliferation rate of 149.56% and 195.50% in MDCK and Vero cells, respectively, compared with that in serum-free medium (SFM). Notably, DS and SP extracts significantly increased superoxide dismutase (SOD) activity, which was 118.61% in MDCK cells and 130.08% in Vero cells for DS, and 108.72% in MDCK cells and 125.63% in Vero cells for SP, indicating a reduction in intracellular oxidative stress. Marine microalgal extracts, especially DS and SP, are feasible alternatives to FBS in cell culture as they promote cell proliferation, ensure safety, and supply essential nutrients while reducing oxidative stress.

## 1. Introduction

Advances in cell culture techniques for growing viruses have led to significant advances in the development of viral vaccines [1,2]. With the increasing demand for vaccines worldwide, developing new techniques to produce vaccines in large quantities has become critical [3]. Several cell lines have been approved for cell culture-based viral vaccine production, including Madin-Darby canine kidney (MDCK) and Vero (African green monkey kidney) cells [4,5]. MDCK cells are particularly useful for influenza virus research and vaccine development, because they are sensitive to almost all types of influenza viruses. Vero cells are susceptible to a wide range of viruses, including polio, hepatitis A, influenza, rabies, and yellow fever viruses [6,7].

Traditionally, animal cells are cultured in a basal medium supplemented with fetal bovine serum (FBS), which is essential for cell growth, including growth factors, hormones, vitamins, and proteins such as fibronectin and albumin [8,9]. However, FBS production requires the extraction of fetal calves from pregnant cows in slaughterhouses, raising serious ethical concerns [10]. In addition, viral, bacterial, and fungal contamination of bovine serum is a major problem in the manufacture of cell culture-based vaccines [11,12], highlighting the need to find serum alternatives such as serum substitutes or serum-free media (SFM).

Recently, notable progress has been made in the development of serum-free cell culture methods. Researchers are working to create inexpensive serum-free media by optimizing the concentration of growth factors and exploring cost-effective options [13]. For example, Kolkman et al. developed SFM for bovine myoblasts, which included supplements such as L-ascorbic acid 2-phosphate, fibronectin, hydrocortisone, Glutamax™, albumin, ITS-X, hIL-6, and α-linolenic acid, as well as growth factors such as fibroblast growth factor 2 (FGF-2), vascular endothelial growth factor (VEGF), insulin-like growth factor 1 (IGF-1), hepatocyte growth factor (HGF), and Platelet-derived growth factor (PDGF)-BB [14]. Li et al. also demonstrated that an optimized growth factor cocktail comprising FGF-2, transforming growth factor (TGF)-β3, and IGF-1 promoted the proliferation and differentiation of human bone marrow-derived mesenchymal stem cells (hMSCs) [15].

Marine microalgae are emerging as a promising alternative because of their high protein, lipid, vitamin, and other nutrient content. Recent studies have highlighted microalgae as valuable sources of pharmacologically active substances [16]. For example, Okamoto et al. [17] reported that microalgal extracts from *Chlorococcus litoralis*, *Stichococcus*, *Chlorella vulgaris*, and *Euglena gracilis* can replace certain nutrients in the media for mammalian cell culture. Chung et al. [18] developed an FBS replacement medium for animal cells using *Spirulina maxima* and found that a *Spirulina vulgaris*-supplemented SFM extract improved cell viability and growth in an embryonic tracheal (EBTr) fibroblast cell line [19].

This study aimed to evaluate the feasibility of using serum-free media by assessing the cell activation efficacy of a marine microalgal extract and growth factor cocktail in MDCK and Vero cells, which are essential for vaccine production.

## 2. Results

### 2.1. Amino Acid, Fatty Acid, and Monosaccaride Composition of Microalgae Extraction

As shown in Table 1, the most abundant amino acids in all species were glutamic acid and aspartic acid, whereas ornithine and citrulline were present in the lowest amounts. Glutamic acid was the most common component in DS (35.06%), SP (17.49%), and NS (15.23%). The highest fatty acid content was observed for palmitic acid in the following order: DS > SP > NS > HP > TS. In particular, the DS contents of palmitic acid, eicosapentaenoic acid (EPA), and myristic acid were 69.89%, 10.90%, and 5.81%, respectively. Among the monosaccharides, NS had the highest content (88.13%), followed by SP (63.16%), and DS (51.68%). Glucose contents were as follows: TS (33.27%), SP (28.98%), and DS (25.88%).

### 2.2. Contamination Detection of Microalgae Extraction

Polymerase chain reaction (PCR) results for the microalgal extract were negative for Mycoplasma and bacteria. The endotoxin levels of microalgae extracts were SP (0.56 EU/mL), DS (0.13 EU/mL), HP (0.10 EU/mL), NS (1.09 EU/mL), and TS (1.50 EU/mL) (Figure 1).

### 2.3. Confirmation of Cytotoxicity of Microalgae Extract and Growth Factor Cocktail

The cytotoxicity of the microalgae extract (50 and 100 µg/mL) from five species and the growth factor cocktail (1 and 10 ng/mL) was confirmed in MDCK and Vero cells (Figure 2 and Figure 3). No cytotoxicity was confirmed in the microalgal extract and growth factor cocktail, and subsequent experiments were conducted in a nontoxic concentration range.

### 2.4. Cell Growth in Microalgae Extract and Growth Factor Cocktail

To analyze the cell growth efficacy of the microalgal extract and growth factor cocktail, cell growth was compared after 24 and 48 h in MDCK and Vero cells (Figure 4 and Figure 5). The cell growth of the microalgae extract and growth factor cocktail was compared with that of OptiPRO^TM^ serum-free medium (SFM, Gibco, Waltham, MA, USA).

OptiPRO SFM was designed for several kidney-derived cell lines, including MDBK, MDCK, and Vero. Several studies have reported the application of OptiPRO SFM in cell culture-based virus propagation [11].

In the case of MDCK cells in microalgae extract (50 µg/mL) and growth factor cocktail (1 and 10 ng/mL), no cell growth was observed in comparison to SFM. However, Vero cell growth was greater than that in SFM at 48 h. Notably, all microalgal extracts (50 µg/mL) and growth factor cocktails (10 ng/mL) demonstrated remarkable efficacy in promoting cell growth (Figure 4). The microalgae extract (100 µg/mL) and growth factor cocktail (1 and 10 ng/mL) demonstrated enhanced cellular growth in MDCK cells when compared to the SFM at 48 h. In particular, SP demonstrated superior efficacy in promoting cell growth compared to SFM. Vero cells demonstrated superior cell growth efficacy compared to SFM at both 24 and 48 h (Figure 5).

### 2.5. Cell Proliferation Assay from Microalgae Extract and Growth Factor Cocktail

BrdU is a synthetic thymidine analog that is incorporated into DNA cells during cell division; therefore, it is commonly used for the detection of proliferating cells [20]. The percentage of BrdU-positive cells showed that the microalgae extract (50 µg/mL) and growth factor cocktail (10 ng/mL) increased in MDCK cells compared to SFM in the following order: DS 149.56%, SP 134.64%, HP 126.15%, and NS 103.64%. However, TS showed lower proliferation than SFM at 95.20%, but higher proliferation than when treated with only the growth factor cocktail (Figure 6A). Vero cells proliferated in all extracts compared with SFM. The growth factor cocktail showed a high proliferation of 161.34%, but DS 195.50% and TS 182.11% with the added microalgae extract showed higher proliferation. In particular, when MDCK and Vero cells were compared, high cell proliferation was observed in the DS group (Figure 6B).

### 2.6. Apoptosis Assay from Microalgae Extract and Growth Factor Cocktail

Apoptosis induced by the microalgal extract (50 µg/mL) and growth factor cocktail (10 ng/mL) was confirmed (Figure 7). MDCK cells showed SFM of 7.65%, SP of 3.50%, DS of 2.95%, HP of 3.2%, NS of 3.30%, and TS of 3.20%. When compared to SFM, microalgae extract (50 µg/mL) and growth factor cocktail (10 ng/mL) did not induce apoptosis or death. Vero cells showed SFM of 4.65%, SP of 4.55%, DS of 4.05%, HP of 3.45%, NS of 4.05%, and TS of 4.95%. Microalgae extracts, excluding TS, did not induce more apoptosis or death than SFM.

### 2.7. SOD Activity from Microalgae Extract and Growth Factor Cocktail

Antioxidant defense systems include enzymes such as superoxide dismutase (SOD), catalase (CAT), and glutathione peroxidase (GPx) [21]. Such antioxidant enzymes protect cells by maintaining O_2_^−^ and H_2_O_2_ at low levels [22]. When compared to SFM 100%, SOD activity was DS 118.61%, HP 115.99%, and SP 108.72% in MDCK cells and DS 130.08%, SP 125.63%, and NS 115.18% in Vero cells. These results showed that DS exhibited the highest SOD activity (Figure 8).

## 3. Discussion

In this study, we found that marine microalgal extracts and growth factor cocktails positively affected the proliferation and survival of MDCK and Vero cells. Specifically, *Dunaliella salina* (DS) and *Spirulina platensis* (SP) extracts significantly increased the proliferation of both cell lines. Notably, DS showed a proliferation rate of 149.56% in MDCK cells and 195.50% in Vero cells compared with SFM.

Second, the compositional analysis of the extracts of the five microalgal species showed that they contained high amounts of glutamic and aspartic acids as amino acids, palmitic acid as a fatty acid, and galactose as a monosaccharide.

Third, the marine microalgal extract and growth factor cocktail were safe against Mycoplasma, bacteria, and endotoxin contaminants.

Fourth, treatment of MDCK and Vero cells with the marine microalgae extract and growth factor cocktail resulted in similar levels of cell death as the control, SFM, confirming that the microalgae extract is safe for cells.

Finally, antioxidant enzyme activities were measured, and it was found that DS and SP extracts significantly increased superoxide dismutase (SOD) activity. DS showed 118.61% SOD activity in MDCK cells and 130.08% in Vero cells, whereas SP showed 108.72% SOD activity in MDCK cells and 125.63% in Vero cells, indicating that marine microalgal extracts can effectively reduce intracellular oxidative stress.

The significant increase in the proliferation of MDCK and Vero cells by DS and SP extracts suggests that these marine microalgae contain bioactive compounds that promote cell growth. The proliferation rates of 149.56% in MDCK cells and 195.50% in Vero cells indicated a strong stimulatory effect and likely provided the nutrients or growth factors required to enhance cellular activity. This is consistent with the existing knowledge that microalgae are rich in nutrients, such as essential amino acids, vitamins, and fatty acids [23].

The nutritional value of microalgal extracts is highlighted by their composition, which contains high levels of glutamic and aspartic acids (amino acids), palmitic acid (fatty acid), and galactose (monosaccharide). Glutamic and aspartic acids are involved in protein synthesis and various metabolic pathways, helping support cell growth and maintenance [24]. Palmitic acid plays an important role in the cell membrane structure and function [25], and galactose plays a crucial role in cellular energy production [26]. Collectively, these components contribute to the overall health and proliferation of the cells.

The confirmation that marine microalgal extracts and growth factor cocktails are safe against Mycoplasma, bacteria, and endotoxin contaminants is of great importance for their application in cell culture and vaccine production. This safety profile eliminates concerns about potential contamination, which is a significant issue with animal-derived sera, such as FBS. Obtaining contaminant-free media is essential to maintain the integrity of cell cultures and prevent adverse effects on cell growth and function [18].

Treatment with the marine microalgae extract and growth factor cocktail resulted in similar levels of cell death as the control (SFM), indicating that these extracts did not cause cytotoxic effects. This indicates that the extracts are biocompatible and do not cause apoptosis or necrosis in the cells, which is crucial for their continued use in long-term cell cultures. This biocompatibility further supports the potential of marine microalgal extracts as safe alternatives to FBS.

The significant increase in superoxide dismutase (SOD) activity observed with DS and SP extracts indicated an enhanced intracellular antioxidant defense mechanism [27,28]. SOD is an important enzyme that mitigates oxidative stress by converting superoxide radicals into less harmful molecules [29]. The increase in SOD activity (118.61% in MDCK and 130.08% in Vero cells for DS and 108.72% in MDCK and 125.63% in Vero cells for SP) suggests that the extract protects cells from oxidative damage and promotes cell viability and longevity. These antioxidant properties are particularly useful in cell culture settings, where oxidative stress can impair cell function and survival. Previous studies have also shown a significant increase in SOD levels with increasing concentrations of marine *Spirulina maxima* extract, similar to the ability of the extract to induce oxidative stress adaptation in cells [18].

The strength of this study is that it confirms the feasibility of developing SFMs using marine microalgal extracts, which are free from the ethical, contamination, and high-cost issues associated with conventional FBS. However, it has limitations in that its stability in long-term culture and affordability need to be evaluated.

Future studies should evaluate the long-term cell culture effects of the five microalgal extracts, especially the DS and SP extracts, and verify their potential for use in vaccine production through cell infection experiments against various viral species.

Furthermore, we will elucidate the molecular mechanism of microalgae extract with growth factor cocktail increase the level of SOD in both MDCK and Vero cell lines.

The results of this study suggest the feasibility of developing vaccine-producing cell lines using marine microalgae, and are expected to provide a new and fruitful basis for the utilization of marine microalgae.

## 4. Materials and Methods

### 4.1. Selection of Microalgae Species

Five species of microalgae were used in this study: *Spirulina platensis* (SP), *Dunaliella salina* (DS), *Haematococcus pluvialis* (HP), *Nanochloropsis salina* (NS), and *Tetraselmis* sp (TS). SP was purchased from California Gold Nutrition (Los Angeles, CA, USA), DS from ALFA Chemistry (Long Island, NY, USA), HP from Gaussdream I (Gyeonggi, Republic of Korea), NS from AKI ORGANIC (Los Angeles, CA, USA), and TS from Chloland (Gyeongnam, Republic of Korea).

### 4.2. Extraction of Microalgae

For microalgal extraction, 10 g of microalgae was mixed with 200 mL of distilled water and extracted in a shaking incubator at 90 °C for 6 h. After extraction, centrifugation was performed at 3200 rpm, 4 °C for 10 min, and the supernatant was filtered using filter paper. After filtration, the samples were freeze-dried and used for the experiments.

### 4.3. Amino Acid, Fatty Acid, and Monosaccharide Composition Analysis

For amino acid component analysis, 50 mL of amino acid buffer was added to 1 g of the sample, followed by ultrasonic extraction for 1 h and extraction at room temperature for 1 h. After filtering with a 0.2 µm syringe filter, HPLC analysis was performed (Appendix A). HPLC analysis conditions used an Inno C18 column (4.6 mm × 15.0 mm, 5 µm/YoungJin biochrom, Gyeonggi, Republic of Korea) and a Dionex Ultimated 3000 (Thermo Fisher Scientific, Waltham, MA, USA) HPLC device. The FL detector analyzed at 340 nm/450 nm and 266 nm/305 nm, and the UV detector analyzed at 338 nm and injection volume of 0.5 µL [30].

For fatty acid analysis, 2 mL of methylation mixture [MeOH: Benzene: DMP (2,2-Dimethoxy-propane): H_2_SO_4_ = 39:20:5:2] was shaken with 1 mL of heptane and extracted at 80 °C for 2 h. A certain amount of the supernatant was extracted and analyzed using GC. For GC, Agilent 7890A (Agilent, Santa Clara, CA, USA) was used, and analysis was performed using a DB-23 (Agilent, 60 mm × 0.25 mm × 0.25 µm) column. The detector is FID (280 °C, H_2_ 35, Air 350, He 35 mL/min) with an injection volume of 1 µL [31].

Carbohydrate component analysis was treated with 2M Trifluoroacetic acid (TFA), and the analysis solvent used was 18 mM NaOH and 200 mM NaoH as eluents, and the analysis was conducted under the conditions of a flow rate of 1.0 mL/min and an injection volume of 20 µL. Measurements were performed using a high-performance anion exchange chromatography-pulsed amperometric detection system (Dionex, Santa Clara, CA, USA) equipped with a CarboPacTM PA1 column (4 mm × 250 mm) [32].

### 4.4. Contamination Analysis

Mycoplasma and bacterial tests were conducted [33]. For Mycoplasma, the e-Myco Mycoplasma PCR Detection Kit (Ver 2.0) (LiliF Diagnostics, Cat No. 25235) was used. For bacteria, the Bacterial 16S rDNA PCR Kit Fast (800) (TaKaRa, Cat No. RR182A) was used to determine microalgal contamination. In addition, an endotoxin test was conducted using a Chromogenic Endotoxin Quant Kit (Thermo Fisher Scientific Inc., A39553).

### 4.5. Cell Culture

MDCK and Vero cells were purchased from Korean Cell Line Bank (Seoul, Republic of Korea). MDCK cells grown in DEME medium (Gibco, Waltham, MA, USA) and Vero cells grown in RPMI 1640 (Gibco, Waltham, MA, USA) containing 10% fetal bovine serum and 1% penicillin, and cultured 37 °C and under 5% CO_2_.

### 4.6. Cell Cytotoxicity

To confirm cytotoxicity, 190 µL of MDCK cells (0.6 × 10^5^ cells/mL) and Vero cells (0.5 × 10^5^ cells/mL) were cultured in each well of a 96-well plate for 24 h. Then, 10 µL of microalgae extract (50 and 100 µg/mL) and growth factor cocktail (TGF-β1; transforming growth factor-β1, PDGF-BB; platelet-derived growth factor-BB, FGF-2; fibroblast growth factor-2, IGF-I; insulin-like growth factor-I, EGF; epidermal growth factor, ITS; insulin transferrin sodium selenite supplement, 1 and 10 ng/mL) were treated. After 24 h, CCK-8 (Cell Counting Kit—8, Dojindo) solution was added, and the absorbance was measured at 450 nm using a microplate reader (BioTek Synergy HT; Agilent Technologies, Santa Clara, CA, USA).

### 4.7. Cell Growth

To confirm the cell growth of the microalgae extract (50 and 100 µg/mL) and growth factor cocktail (1 and 10 ng/mL), cell growth was confirmed for 24 and 48 h in MDCK and Vero cells. After culturing the cells in 96 wells, the medium was removed 24 h later, the growth factor cocktail (1 and 10 ng/mL) was mixed with FBS-free medium, 190 µL was dispensed, and the remaining 10 µL was cultured with each microalgae extract. Then, 10 μL of CCK-8 solution was treated to check cell growth for 24 and 48 h.

### 4.8. Cell Proliferation Assay

BrdU (5-Bromo-2′-deoxyuridine) analysis was performed to confirm cell proliferation [34]. MDCK and Vero cells were cultured on slides for 24 h. The medium without FBS was replaced with medium containing 10 ng/mL of growth factor cocktail, and 50 µg/mL of microalgae extract was added and cultured for 24 h. The 5-Bromo-2′-deoxyuridine Labeling and Detection Kit Ⅰ (Roche, Basel, Swiss) was used, and cell nuclei were stained using Hoechst 33342 (Thermo Scientific, Waltham, MA, USA) fluorescent dye and observed using a fluorescence microscope (Nikon Eclipse 80i, Nikon, Tokyo, Japan). Quantitative data expressing BrdU-positive cells (green) were assessed by counting positively stained cells using ImageJ software version 1.54d (National Institutes of Health, Bethesda, MD, USA).

### 4.9. Apoptosis Assay

MDCK and Vero cells were cultured in 60 mm dish for 24 h. The medium without FBS was replaced with medium containing 10 ng/mL of growth factor cocktail, and 50 µg/mL of microalgae extract was added and cultured for 24 h. Apoptosis was measured using the FITC Annexin V Apoptosis Detection Kit II (BD Biosciences, San Diego, CA, USA). Apoptosis was measured by flow cytometry (BD Biosciences).

### 4.10. Superoxide Dismutase (SOD) Activity

The SOD activity was measured using an EZ-SOD assay kit (Dogenbio Co., Seoul, Republic of Korea). MDCK and Vero cells were cultured on a six-well plate for 24 h. The medium without FBS was replaced with medium containing 10 ng/mL of growth factor cocktail, and 50 µg/mL of microalgae extract was added and cultured for 24 h. Next, the culture medium was removed, cells were washed with PBS, proteins were extracted using RIPA buffer (Rockland Immunochemical, Limerick, PA, USA), and proteins were quantified using a BCATM protein analysis kit (Thermo Scientific, Waltham, MA, USA). Next, 20 μL each was added to the sample well and Blank2 well, and 20 μL of distilled water was added to each well of Blank1 and Blank3. Then 200 μL of WST working solution was added to each well, and 20 μL of dilution buffer was added to each well of Blank2 and Blank3. After adding 20 μL of enzyme working solution to Blank1 and sample well, the plate was reacted at 37 °C for 20 min. The absorbance was measured at 450 nm using a plate reader.

### 4.11. Statistical Analysis

All experiments were repeated three times, and the data are expressed as mean ± standard deviation. Statistical analyses were performed using the GraphPad Prism software version 10.2.3 (San Diego, CA, USA). One-way ANOVA was performed to verify the significance of each sample, and a post hoc test was performed using Dunnett’s multiple comparison test (*p* < 0.05).

## 5. Conclusions

Marine microalgal extracts and growth factor cocktails promoted cell proliferation and antioxidant activity in MDCK and Vero cells. In particular, DS and SP extracts exhibited the potential to replace FBS. This study provides an alternative to address the ethical issues and contamination risks of FBS, and may contribute to increasing the efficiency and safety of future vaccine production processes. Further studies are required to validate the practicality of marine microalgal extracts and expand their applicability to different cell lines and viral species.

## Figures and Tables

**Figure 1 ijms-25-09881-f001:**
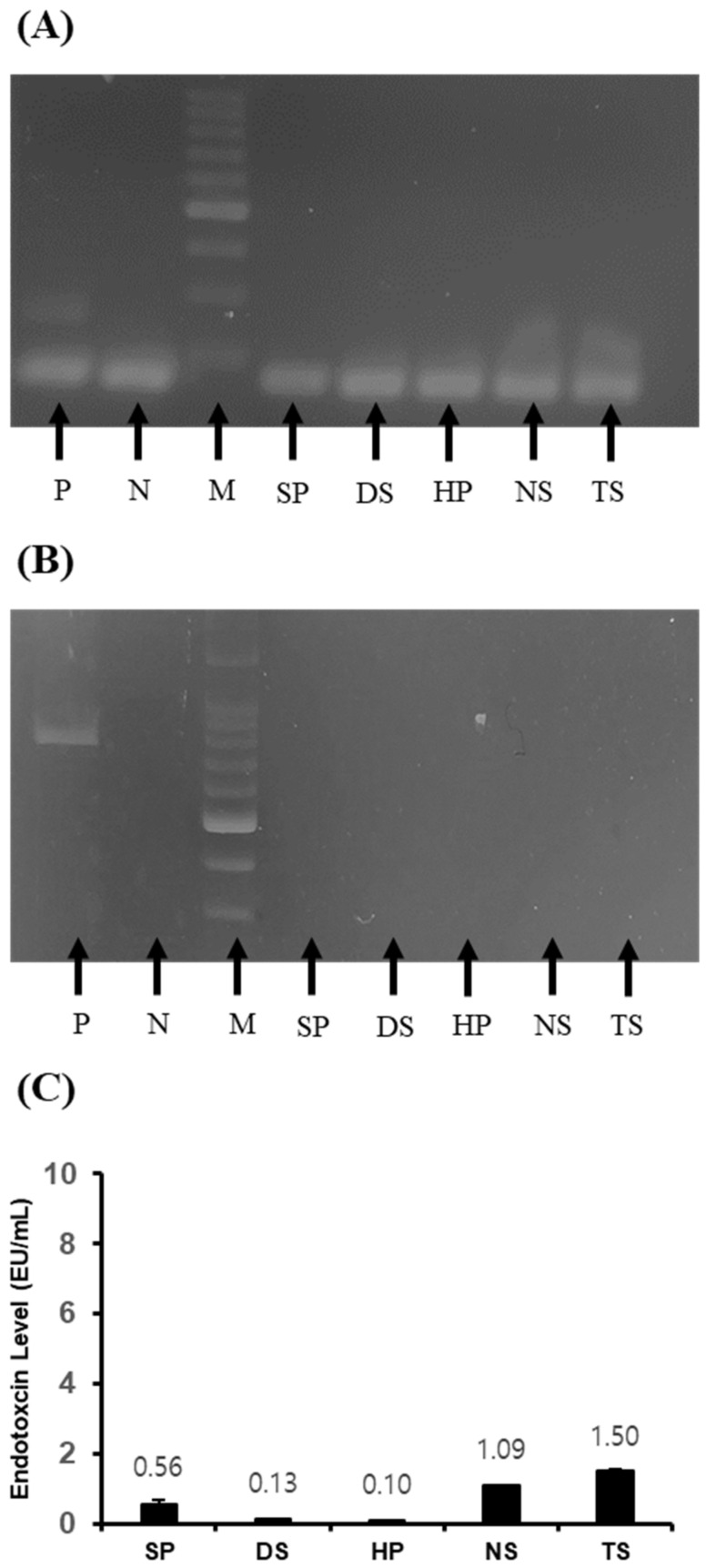
Contamination detection in the microalgae extraction. (**A**) Mycoplasma, (**B**) bacteria, and (**C**) endotoxin. P: positive, N: negative, M: marker, SP: *Spirulina plantensis*, DS: *Dunaliella salina*, HP: *Haematococcus pluvialis*, NS: *Nannochloropsis salina*, and TS: *Tetraselmis* sp.

**Figure 2 ijms-25-09881-f002:**
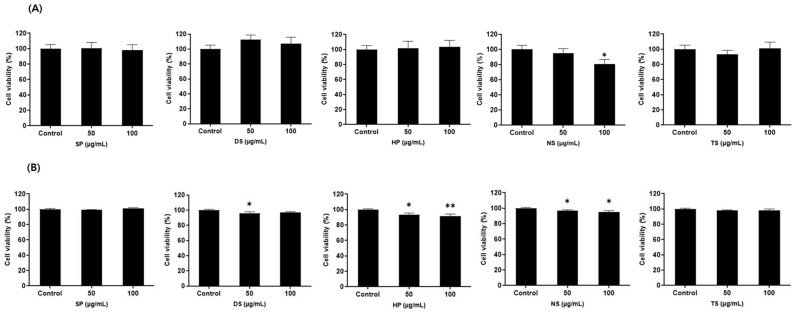
Microalgae extract (50 and 100 µg/mL) cytotoxicity in Madin-Darby canine kidney (MDCK) and Vero cells. (**A**) MDCK and (**B**) Vero cells. Values are expressed as mean ± standard deviation (SD) from three replications. * *p* < 0.05, ** *p* < 0.01 indicate values compared to serum-free medium (SFM). SP: *Spirulina plantensis*, DS: *Dunaliella salina*, HP: *Haematococcus pluvialis*, NS: *Nannochloropsis salina*, and TS: *Tetraselmis* sp.

**Figure 3 ijms-25-09881-f003:**
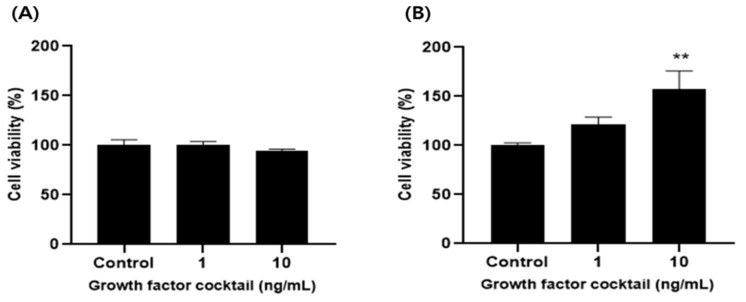
Growth factor cocktail (1 and 10 ng/mL) cytotoxicity in Madin-Darby canine kidney (MDCK) and Vero cells. (**A**) MDCK and (**B**) Vero cells. Values are expressed as mean ± standard deviation (SD) from three replications. ** *p* < 0.01 indicate values compared to serum-free medium (SFM).

**Figure 4 ijms-25-09881-f004:**
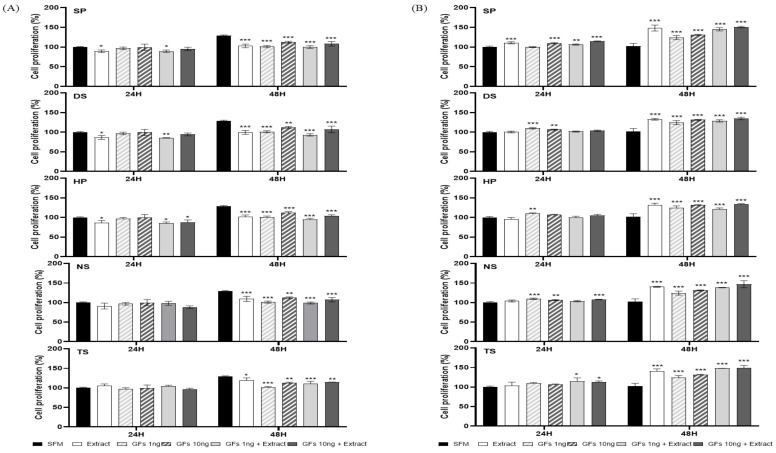
Madin-Darby canine kidney (MDCK) and Vero cell growth with microalgae extract (50 µg/mL) and growth factor cocktail. (**A**) MDCK and (**B**) Vero cells. Values are expressed as mean ± standard deviation (SD) from three replications. * *p* < 0.05, ** *p* < 0.01, and *** *p* < 0.001 indicate values compared to serum-free medium (SFM). SP: *Spirulina plantensis*, DS: *Dunaliella salina*, HP: *Haematococcus pluvialis*, NS: *Nannochloropsis salina*, and TS: *Tetraselmis* sp.

**Figure 5 ijms-25-09881-f005:**
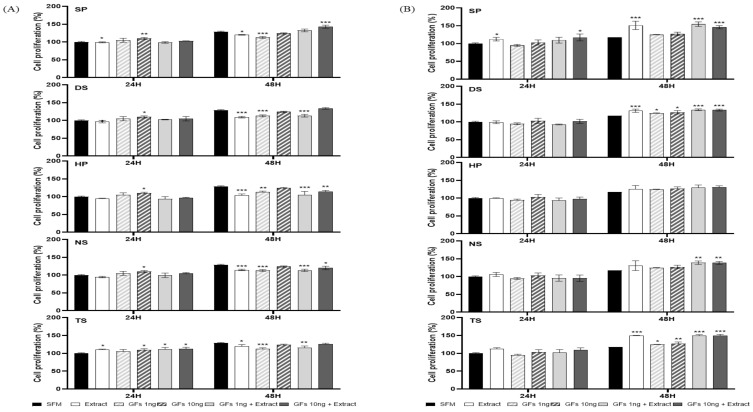
Madin-Darby canine kidney (MDCK) and Vero cell growth with microalgae extract (100 µg/mL) and growth factor cocktail. (**A**) MDCK and (**B**) Vero cells. Values are expressed as mean ± standard deviation (SD) from three replications. * *p* < 0.05, ** *p* < 0.01, and *** *p* < 0.001 indicate values compared to serum-free medium (SFM). SP: *Spirulina plantensis*, DS: *Dunaliella salina*, HP: *Haematococcus pluvialis*, NS: *Nannochloropsis salina*, and TS: *Tetraselmis* sp.

**Figure 6 ijms-25-09881-f006:**
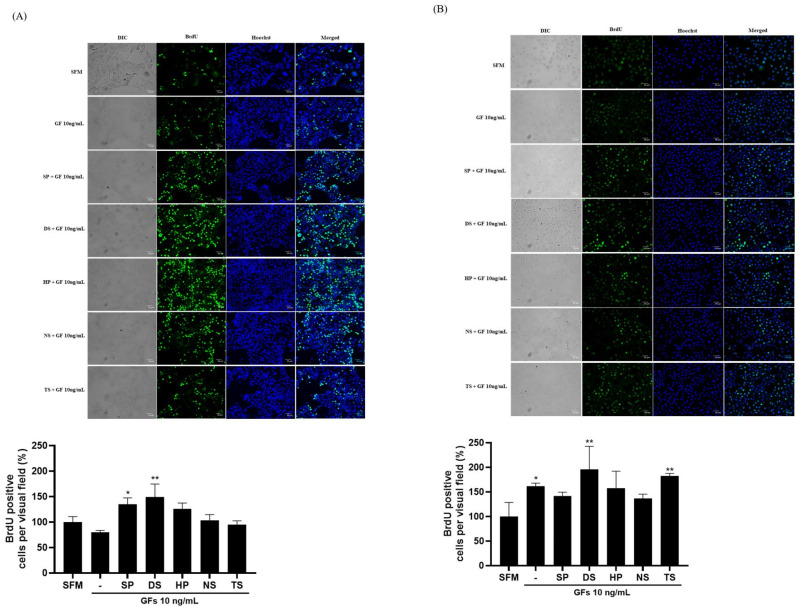
Madin-Darby canine kidney (MDCK) and Vero cell growth with microalgae extract (50 µg/mL) and growth factor cocktail. Scale bar = 50 µm. (**A**) MDCK and (**B**) Vero cells. Values are expressed as mean ± standard deviation (SD) from three replications. * *p* < 0.05 and ** *p* < 0.01 indicate values compared to serum-free medium (SFM). SP: *Spirulina plantensis*, DS: *Dunaliella salina*, HP: *Haematococcus pluvialis*, NS: *Nannochloropsis salina*, TS: *Tetraselmis* sp.

**Figure 7 ijms-25-09881-f007:**
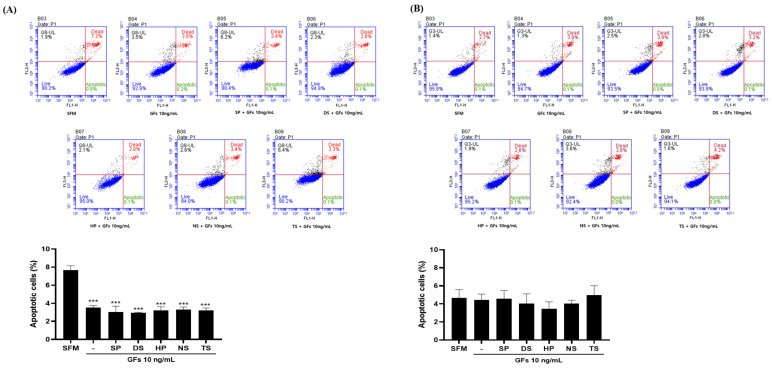
Madin-Darby canine kidney (MDCK) and Vero apoptosis assay cultured in microalgae extract (50 µg/mL) and growth factor cocktail (10 ng/mL). (**A**) MDCK and (**B**) Vero cells. Values are expressed as mean ± standard deviation (SD) from three replications. *** *p* < 0.001 indicate values compared to serum-free medium (SFM). SP: *Spirulina plantensis*, DS: *Dunaliella salina*, HP: *Haematococcus pluvialis*, NS: *Nannochloropsis salina*, TS: *Tetraselmis* sp.

**Figure 8 ijms-25-09881-f008:**
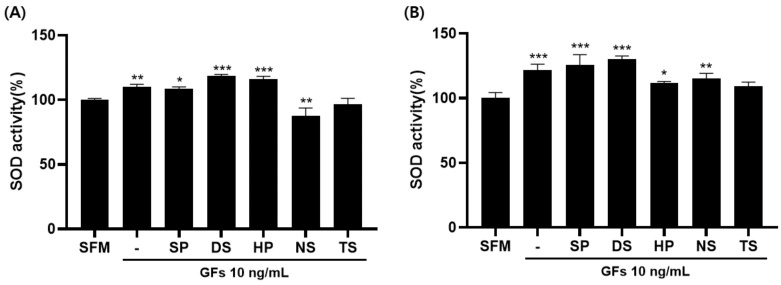
Superoxide dismutase (SOD) activity of microalgae extract and growth factor cocktail. (**A**) Madin-Darby canine kidney (MDCK) and (**B**) Vero cells. Values are expressed as mean ± standard deviation (SD) from three replications. * *p* < 0.05, ** *p* < 0.01, and *** *p* < 0.001 indicate values compared to serum-free medium (SFM). SP: *Spirulina plantensis*, DS: *Dunaliella salina*, HP: *Haematococcus pluvialis*, NS: *Nannochloropsis salina*, TS: *Tetraselmis* sp.

**Table 1 ijms-25-09881-t001:** Amino acid, fatty acid, and monosaccaride composition of microalgae extraction.

		Concentration (%)				
		*Spirulina platensis*	*Dunaliella salina*	*Haematococcus pluvialis*	*Nannochloropsis salina*	*Tetraselmis* sp.
**Amino acids**	Glutamic acid	17.49	35.06	13.16	15.23	12.38
	Aspartic acid	12.66	10.51	10.34	13.85	10.7
	Alanine	8.11	0.28	7.25	7.53	6.98
	Leucine	8.46	6.91	7.74	9.01	8.04
	Serine	5.55	4.14	5.62	5.27	6.12
	Glycine	6.10	4.30	6.10	6.28	5.86
	Threonine	6.10	4.99	5.55	5.37	5.37
	Arginine	5.02	6.20	5.65	6.40	7.79
	Valine	6.27	5.06	6.34	6.48	6.10
	Phenyalanine	5.23	4.66	5.52	5.67	5.38
	Isoleucine	5.00	4.33	5.14	5.42	5.32
	Lysine	4.96	3.04	4.60	3.69	5.60
	Tyrosine	2.45	2.59	2.49	3.37	3.20
	Proline	5.08	3.62	4.23	4.57	4.78
	Histidine	1.52	4.32	1.20	1.64	1.56
	Taurine	0.00	0.00	5.94	0.00	1.16
	GABA	0.00	0.00	0.00	0.00	3.27
	Ornitnine	0.00	0.00	2.77	0.22	0.27
	Citrulline	0.00	0.00	0.36	0.00	0.11
**Fatty acids**	Palmitic acid	67.84	69.89	48.74	64.13	24.87
	Arachidic acid	0.00	0.00	1.10	0.00	1.84
	Myristic acid	5.63	5.81	3.85	8.96	7.66
	Stearic acid	2.16	3.60	2.90	5.44	1.02
	Lignoceric acid	0.00	0.00	1.94	0.00	1.00
	Palmitoleic acid	0.00	0.00	1.92	0.00	0.68
	Behenic acid	0.00	0.00	9.01	0.00	0.00
	Lauric acid	0.00	0.00	1.40	0.00	0.00
	Capric acid	0.00	0.00	0.00	1.47	0.00
	Arachidonic acid	13.05	4.19	2.00	0.00	12.15
	Oleic acid	9.59	3.55	11.18	18.51	14.05
	Gamma-linolenic acid	0.00	0.00	0.85	0.00	1.06
	Alpha-linolenic acid	0.00	0.00	6.19	0.00	9.71
	Linoleic acid	0.86	2.05	8.08	1.49	7.16
	Dihomo-gamma-linolenic acid	0.00	0.00	0.00	0.00	0.65
	Eicosapentaenoic acid	0.88	10.90	0.82	0.00	18.16
**Monosaccaride**	Galactose	63.16	51.68	19.52	88.13	23.45
	Glucose	28.98	25.88	14.08	2.32	33.27
	Fucose	5.49	3.82	3.23	5.53	22.37
	Arabinose	0.00	3.95	3.47	0.00	0.64
	Rhamnose	0.00	0.00	0.40	0.00	0.00
	N.D.	2.37	14.67	59.31	4.03	20.27

N.D.: Non determined.

## Data Availability

Data are contained within the article.

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
