# Peer review of "Serum-Free Media Formulation Using Marine Microalgae Extracts and Growth Factor Cocktails for Madin-Darby Canine Kidney and Vero Cell Cultures"

_ijms, 2024, doi:10.3390/ijms25189881_

Round 1

Reviewer 1 Report

Comments and Suggestions for Authors

The authors studied the effects of several marine microalgae extracts and growth factor cocktails on cell proliferation, which provided excellent guidance for the development of serum-free media. The experimental design of the entire manuscript is reasonable and the results are clearly explained. But there are still some minor problems that need to be solved.

1. The authors should add a normal medium group (including FBS) as positive control to increase the validity of the experimental results

2. In Figure 1, the author should compare whether there are differences in various components in different marine microalgae extracts and perform appropriate statistical analysis, marking significance and error bars.

3. Figure 2 has the same problem as Figure 1, and the result does not have error bars. There is no indication of the number of experiments in the figure legend.

4. In Figure 4A, growth factors seem to have different effects in the two cells. Can the author explain why?

5. The authors should explain why these two types of cells were chosen as experimental subjects, whether they are representative?

6. The author detected the main components in the extract, which was very good. So what do the authors think are the main components in the extract that play a role in replacing FBS?

Comments on the Quality of English Language

“已经过社区验证”图标

no additional comments

Reviewer 2 Report

Comments and Suggestions for Authors

This manuscript reported biological assessments of five marine microalgal extracts using MDCK and Vero cells. The result revealed that DS and SP extracts exhibited the potential to replace FBS. This study provides an alternative to address the ethical issues and contamination risks of FBS, and may contribute to increasing the efficiency and safety of future vaccine production processes. Based on these findings, this work is suggested to be published in the forthcoming issue of this journal.

However, there were some issues to be addressed as followings:

1. Have the cytotoxicity and SOD activity of these marine microalgae extracts been studied? What was the aim to use these microalgae extracts? These should be mentioned in the Introduction.

2. What was the purpose to analysis amino acid, fatty acid, and monosaccaride composition of microalgae extraction? Usually, the relationship between the contents of these compositions and the biological activities should be assessed.

3. As shown in Figure 3, the cell viability increased with the increase of concentration for some algae extracts such as HP and TS. Whereas, it decreased for some algae extracts such as DS and NP. Why? Notably, the label ‘NP’ in Figure 3A should be revised as ‘NS’. Please check it.

Others:

1. Figure 1 was a bit blurry.

2. Please pay attentions to the subscript fonts for the numbers in the formulas H2SO4 and H2.

Reviewer 3 Report

Comments and Suggestions for Authors

The present manuscript deals with the potential use of marine microalgal extracts to develop serum-free media as an alternative to fetal bovine serum. Works like that are important for the development of cell culture techniques with regard to safety and efficacy.

The experimental design is appropriate to test the hypothesis and the figures, diagrams, and schemes are also suitable. The conclusions are consistent with the evidence and arguments presented; and the references are relevant. The entire study and the manuscript are of a very high standard and practically no comments can be made. I can happily agree to publish the manuscript in the same state as it is now presented.

Round 2

Reviewer 1 Report

Comments and Suggestions for Authors

The authors have significantly improved the manuscript,. The overall science is sound. I have no additional concerns or comments.